# Long-Term Outcomes of Ledipasvir/Sofosbuvir Treatment in Hepatitis C: Viral Suppression, Hepatocellular Carcinoma, and Mortality in Mongolia

**DOI:** 10.3390/v17060743

**Published:** 2025-05-22

**Authors:** Amgalan Byambasuren, Buyankhishig Gyarvuulkhasuren, Byambatsogt Erdenebat, Khurelbaatar Nyamdavaa, Oidov Baatarkhuu

**Affiliations:** 1Department of Health Research, Graduate School, Mongolian National University of Medical Sciences, Ulaanbaatar 14210, Mongolia; amgalan.ara@gmail.com; 2General Hospital of the Arkhangai Province, Tsetserleg 65191, Mongolia; buyaamgl24@gmail.com; 3Health Department of the Arkhangai Province, Tsetserleg 65089, Mongolia; byambaa0193@gmail.com; 4Department of Physiology, Mongolian National University of Medical Sciences, Ulaanbaatar 14210, Mongolia; khurelbaatar.n@mnums.edu.mn; 5Department of Infectious Diseases, School of Medicine, Mongolian National University of Medical Sciences, Ulaanbaatar 14210, Mongolia; 6Mongolian Academy of Medical Sciences, Ulaanbaatar 14210, Mongolia

**Keywords:** viral suppression, liver cirrhosis, chronic liver disease, antiviral therapy

## Abstract

(1) Background: Hepatitis C virus (HCV) infection poses a significant health burden, particularly in Mongolia, where the HCV prevalence is notably high. This study evaluates the long-term outcomes of HCV treatment with ledipasvir/sofosbuvir, focusing on mortality, viral relapse, and hepatocellular carcinoma (HCC) development. (2) Methods: This prospective, longitudinal cohort study initially enrolled patients with chronic HCV in Mongolia between 2016 and 2017, focusing on those who completed the five-year follow-up (*n* = 303). The study measured long-term mortality, HCC development, and viral relapse, employing non-invasive methods to assess liver fibrosis and liver function. (3) Results: At the outset, 98.2% of the patients achieved undetectable HCV RNA levels. Over five years, 6.27% experienced viral relapse and 3.30% developed hepatocellular carcinoma (HCC), with a mortality rate of 5.94%. In a multivariable analysis, the significant predictors for HCC occurrence included age (OR = 1.081, 95% CI = 1.021–1.145), liver cirrhosis (OR = 5.866, 95% CI = 1.672–22.577), and GGT level (OR = 1.011, 95% CI = 1.004–1.018). The independent predictors of mortality included age (OR = 1.083, 95% CI = 1.024–1.147), liver cirrhosis (OR = 6.529, 95% CI = 1.913–22.281), and GGT (OR = 1.011, 95% CI = 1.004–1.017). (4) Conclusions: This study demonstrates that ledipasvir/sofosbuvir effectively suppresses HCV initially and maintains low viral relapse rates over the long term. However, it emphasizes the need for continued management to reduce the long-term risk of HCC and mortality, especially in patients with severe liver fibrosis or cirrhosis.

## 1. Introduction

Hepatitis C virus (HCV) infection remains a global health challenge, particularly in regions such as Mongolia, where the seroprevalence is notably high [1,2,3]. Approximately 6% of the Mongolian population is infected with HCV—a prevalence considerably exceeding global averages [4,5]. This high prevalence has substantial implications for public health, given the disease’s potential progression to chronic liver diseases, including cirrhosis and hepatocellular carcinoma (HCC), which remain leading causes of morbidity and mortality worldwide [1,2,3,5].

The introduction of direct-acting antivirals (DAAs) has significantly improved the treatment outcomes for HCV infection [6,7]. Unlike earlier treatments with pegylated interferon and ribavirin, which showed limited efficacy and considerable side effects, DAAs such as ledipasvir/sofosbuvir have demonstrated high sustained virological response (SVR) rates and improved tolerability [7,8,9]. However, the high cost of DAAs and the need for long-term outcome monitoring present challenges, particularly in high-prevalence settings. Despite these advances, the long-term impact of DAAs on HCV-related complications, particularly the risk of HCC development and the recurrence of the virus, is not fully understood [10,11,12,13,14,15]. A preliminary study conducted in 2020, involving over 5000 Mongolian patients treated with ledipasvir/sofosbuvir, provided initial insights into the efficacy of this regimen. However, it lacked the long-term follow-up data essential for assessing the durability of the treatment and the evolution of liver disease post-therapy [9].

To address this gap, the present study evaluates the five-year clinical trajectories of these patients following treatment. Specifically, it assesses the incidence of viral relapse, mortality, and new cases of HCC, thus providing a comprehensive view of the long-term effectiveness of ledipasvir/sofosbuvir in a high-prevalence region. The outcomes of this research are expected to influence treatment protocols, policy decisions, and patient management strategies in Mongolia and similar contexts globally.

## 2. Materials and Methods

### 2.1. Study Design and Patient Follow-Up

Initially, 1203 patients diagnosed with chronic hepatitis C virus (CHC) were identified as eligible after they successfully completed a 12-week treatment regimen with ledipasvir/sofosbuvir and achieved a sustained virological response (SVR) at 24 weeks post-treatment. Of these, 505 agreed to enroll in the extended study and provided informed consent, establishing our study baseline. From this group, 303 participants successfully completed the five-year follow-up (Figure 1).

Participants were excluded from the final analysis for several reasons: those who were lost to follow-up, were older than 80 years, had any organ end-stage disorders, withdrew consent during the study, failed to comply with the study’s requirements, or had incomplete data records that precluded comprehensive outcome assessment.

The study was conducted following the ethical standards of the Declaration of Helsinki and was approved by the medical ethical committee of the Mongolian National University of Medical Sciences (Approval No: 2022/Z-04, dated 29 April 2022). All participants provided informed consent for both the initial treatment phase and the extended follow-up period, with strict adherence to data privacy and patient confidentiality.

### 2.2. Clinical Measures and Follow-Up Procedures

Cirrhosis was diagnosed based on a combination of clinical signs (e.g., splenomegaly, ascites), imaging findings (e.g., nodular liver contour or splenomegaly on ultrasound), and/or non-invasive fibrosis scores (e.g., FIB-4 > 3.25 or APRI > 1.5), when available.

This longitudinal study instituted rigorous annual follow-up assessments to meticulously monitor the long-term health outcomes of the participants, with an emphasis on the durability of treatment efficacy and the progression of hepatic conditions. Annually, a comprehensive battery of laboratory tests was administered to assess systemic health and specific liver functionality. These evaluations included complete blood counts to identify hematological abnormalities that might indicate infection or anemia, liver function tests such as alanine aminotransferase (ALT) and aspartate aminotransferase (AST) to monitor hepatic enzyme levels, and quantitative HCV RNA testing to detect any viral resurgence indicative of a relapse.

Given the logistical and ethical challenges associated with invasive liver biopsies, particularly in resource-limited rural settings, the study leveraged non-invasive methodologies to estimate hepatic fibrosis. The AST to Platelet Ratio Index (APRI) and the Fibrosis-4 (FIB-4) index were calculated annually as surrogate markers of liver fibrosis, reducing the need for biopsy-based assessment.

In parallel, annual diagnostic ultrasonography was performed as a key surveillance tool for hepatocellular carcinoma (HCC). This imaging was especially targeted at participants exhibiting elevated fibrosis markers or atypical alpha-fetoprotein levels, which are often harbingers of malignancy. The early detection of HCC via ultrasonography is crucial, as it substantially improves treatment outcomes and survival rates.

The study’s primary endpoints—mortality, viral relapse, and the development of HCC—were chosen to comprehensively assess the long-term impact of ledipasvir/sofosbuvir therapy. These outcomes reflect not only the antiviral therapy’s effects on viral suppression and survival but also its role in preventing progression to advanced liver disease.

### 2.3. Treatment Regimen and Monitoring

The foundational treatment regimen for this cohort study was a 12-week course of ledipasvir/sofosbuvir, a combination known for its efficacy in treating hepatitis C [16,17]. This direct-acting antiviral therapy is well-documented for its ability to achieve high rates of sustained virological response, particularly among patients with HCV genotypes 1a or 1b, which were prevalent in our study population. Virological response was defined as a sustained virological response at 24 weeks post-treatment (SVR24), indicated by undetectable HCV RNA levels measured 24 weeks after the end of therapy.

Throughout the subsequent five-year observational period, the standard protocol did not include the administration of additional antiviral treatments unless specific conditions were met. Monitoring for HCV relapse was an integral part of the follow-up regimen, involving regular and systematic viral load testing. If a relapse was detected—characterized by the reappearance of HCV RNA in a patient previously confirmed as having achieved a sustained virological response—intervention protocols were promptly initiated.

Retreatment decisions were made in accordance with the most current clinical guidelines and drug protocols approved by national health authorities, ensuring that the therapeutic interventions remained up-to-date and aligned with the evolving landscape of hepatitis C management [16,17,18,19,20]. This approach not only ensured adherence to best medical practices but also allowed the study to adapt to the new advancements in treatment options that emerged over the course of the follow-up period.

In addition to monitoring for viral relapse, overall health status and liver function were assessed through regular clinical evaluations and laboratory investigations. This comprehensive monitoring strategy was essential for the timely detection of potential complications related to the disease or its treatment and for making appropriate adjustments to patient management.

### 2.4. Statistical Analysis

Descriptive statistics were used to summarize the patient characteristics at baseline and during follow-up, with the categorical variables compared using chi-square tests and the continuous variables analyzed using independent t-tests or Mann–Whitney U tests, as appropriate. The longitudinal changes in liver function markers, fibrosis scores, and viral load were analyzed using repeated measures ANOVA to account for within-subject variability over time.

Cox proportional hazards regression models were employed to identify the independent predictors of these long-term outcomes. The model assessed the potential risk factors associated with relapse, mortality, and HCC development, adjusting for relevant covariates such as baseline liver function, fibrosis status, and demographic characteristics. Odds ratios (OR) with corresponding 95% confidence intervals (CI) were calculated to quantify the relative risk of adverse outcomes over time. To avoid multicollinearity, variables that are components of composite scores such as FIB-4 (e.g., age, AST, ALT, and platelet count) were not included simultaneously in the multivariable models; instead, FIB-4 was retained as the representative marker of liver fibrosis.

All statistical analyses were performed using SPSS software, version 27.0, ensuring methodological rigor and adherence to standard analytical practices in clinical epidemiology. A significance threshold of *p* < 0.05 was applied to determine statistical significance.

## 3. Results

The cohort analysis included 303 patients, after excluding those lost to follow-up, to establish the baseline. A comparison of the baseline characteristics between the participants retained in follow-up (*n* = 303) and those lost to follow-up (*n* = 202) showed no major differences in sex, liver cirrhosis status, or key laboratory markers such as ALT, AST, platelet count, APRI, and FIB-4 scores (Appendix A). Although age differed slightly between the groups (median 52.4 vs. 51.9 years, *p* = 0.003), the absolute difference was minimal and not considered clinically significant.

The mean age of the study participants was 53.4 years, with a 95% confidence interval ranging from 52.0 to 54.7 years. Of these participants, 34.2% (*n* = 103) were male, and the mean BMI was recorded at 26.5 kg/m^2^. Regarding liver health, 17.6% of the patients (*n* = 53) presented with liver cirrhosis, while 4.3% (*n* = 13) had experience with interferon treatment. The baseline laboratory tests revealed a mean ALT level of 30.1 U/L (CI: 28.1–32.5 U/L), AST level of 29.2 U/L (CI: 27.2–31.3 U/L), total bilirubin of 0.63 mg/dL (CI: 0.60–0.67 mg/dL), GGT level of 49.3 U/L (CI: 44.6–53.9 U/L), ALP of 98 U/L (CI: 93.2–102.9 U/L), and total albumin level at 42.1 g/L (CI: 42–49.2 g/L). The platelet counts were 204.6 × 10^9^/L (CI: 191–218 × 10^9^/L), and the HCV RNA levels were notably variable with a mean of 18,006.6 IU/mL and a standard deviation of 231,196.9 IU/mL. The APRI score was calculated at 0.48 (CI: 0.42–0.55), categorizing 78.1% of the participants as F0 (>0.5), 18.3% as F1–2 (0.51–1.5), and 3.7% as F3–4 (>1.51). Similarly, the mean FIB-4 score was 1.53 (CI: 1.49–1.68), classifying 65.1% of patients as F0 (>1.45), 28.6% as F1–2 (1.46–3.25), and 6.3% as F3–4 (>3.25).

Following the initial treatment, 98.2% of our cohort (297 out of 303 patients) exhibited no detectable HCV RNA at baseline, indicating effective viral suppression. During the follow-up, 16 patients (5.3%) experienced a relapse of HCV infection, as confirmed by detectable HCV RNA demonstrating the persistence of viral activity. The stability of non-detectable HCV RNA was maintained throughout the follow-up period, as evidenced in Figure 2.

However, the follow-up also uncovered ongoing challenges with HCC development and mortality, which occurred across participants regardless of their HCV RNA status. This suggests that factors beyond viral presence, such as liver fibrosis or genetic predisposition, may influence these outcomes. Specifically, HCC developed in 10 patients (3.30%), and 18 patients (5.94%) died by the study’s end. Notably, by the fifth year, while 96.0% of those with non-detected HCV RNA survived without HCC, the proportion of participants who developed HCC or died increased from 1.0% at baseline to 5.4% at the end of the follow-up (Table 1). This trend indicates that additional factors might contribute to disease progression and mortality, highlighting the complex nature of HCV infection management.

The association between the clinical outcomes and the baseline characteristics of the patients is detailed in Table 2, Table 3 and Table 4. The analyses reveal that the patients who developed HCC or died exhibited notably poorer liver health at baseline. Specifically, a significantly higher percentage of patients who developed HCC had liver cirrhosis (73.7%) compared to those who did not develop HCC (13.4%). This association underscores the role of liver cirrhosis as a major risk factor for severe outcomes. Furthermore, the baseline liver function tests were significantly worse in the patients with adverse outcomes. The mean AST level in the patients who developed HCC was 45.7 U/L, markedly higher than the 27.7 U/L observed in the patients without HCC.

Similarly, ALT levels were elevated at 41.5 U/L in the HCC patients compared to 26.9 U/L in the non-HCC group, with both results reaching statistical significance (*p* = 0.001). In addition, the scores used to assess liver fibrosis, APRI, and FIB-4, were considerably higher in the patients who developed complications. The median FIB-4 score among the patients with HCC was 2.86, reflecting advanced fibrosis, compared to a score of 1.48 in those without HCC. These findings indicate that the severity of liver fibrosis is a critical determinant of outcomes in chronic-HCV-infected patients, correlating with both the development of HCC and overall mortality.

The univariable and multivariable regression analyses identify several key predictors of HCC occurrence (Table 5). Age significantly increases the risk of developing HCC, with the multivariable odds ratio (OR) indicating that each additional year increases the risk by approximately 8% (OR = 1.081, 95% CI = 1.021–1.145, *p* = 0.003). Liver cirrhosis emerges as a strong predictor in both models, increasing the risk of HCC more than fivefold (OR = 5.866, 95% CI = 1.672–22.577, *p* = 0.006). Interferon experience showed a significant association in the univariable model (OR = 5.006, *p* = 0.023), although this was not sustained in the multivariable analysis, suggesting other confounding factors may mitigate this effect. Liver enzymes such as ALT and AST, along with APRI, were identified as significant risk factors in the univariable analysis; however, they were not included in the multivariable model due to the potential collinearity with composite fibrosis indices such as the FIB-4 score. The fibrosis marker FIB-4 also indicated increased risk, which retained significance in the univariable analysis (OR = 2.077, 95% CI = 1.434–3.008, *p* < 0.001).

In terms of mortality (Table 6), age again plays a significant role, with older age substantially increasing the risk (multivariable OR = 1.083, 95% CI = 1.024–1.147, *p* = 0.006). Liver cirrhosis remains a critical predictor, with a more than sixfold increase in the mortality risk (OR = 6.529, 95% CI = 1.913–22.281, *p* = 0.003). GGT remained a significant factor in the multivariable model (OR = 1.011, 95% CI = 1.004–1.017, *p* = 0.002), reinforcing its association with adverse liver-related outcomes. In contrast, FIB-4, although significant in the univariable analysis (OR = 1.636, *p* = 0.001), was not an independent predictor in the adjusted model (*p* = 0.589). Other markers such as ALT, AST, albumin, and APRI score were excluded from the final models due to the collinearity with FIB-4 or a lack of statistical significance (Table 6).

## 4. Discussion

Our cohort study of 303 chronic hepatitis C virus (HCV) patients presents a nuanced understanding of the intricate dynamics among liver fibrosis, HCV RNA detection, and clinical outcomes such as hepatocellular carcinoma (HCC) and overall mortality. Extending the focus beyond the short-term effects common in prior research, our findings offer a deeper insight into the long-term consequences of HCV infection, illustrating the significant but complex benefits of sustained viral suppression and highlighting the ongoing challenges in liver health management over extended periods [21,22,23]. Therefore, our research not only contributes to the existing body of knowledge but also opens several avenues for further scientific inquiry and clinical innovation aimed at improving the life expectancy and quality of life for patients afflicted with HCV.

The initial viral suppression rate observed in our cohort, where 98.2% exhibited non-detectable HCV RNA at baseline, significantly exceeds the rates reported in previous studies, indicating a robust response to the ledipasvir/sofosbuvir regimen [24,25,26,27]. This level of early suppression is promising but must be juxtaposed with the persistent risk of severe liver complications, suggesting that successful treatment and non-detectability of the virus do not confer immunity from future hepatic deterioration. The continuation of non-detectable HCV RNA levels throughout the study reinforces the regimen’s efficacy but also emphasizes the necessity for ongoing vigilance in patient monitoring.

The role of liver cirrhosis as a major risk enhancer for developing HCC, as demonstrated in our study, aligns with the existing literature and confirms the perilous trajectory for patients with significant fibrotic progression [28,29]. The stark increase in risk, with cirrhosis heightening the likelihood of HCC sixfold, urgently calls for aggressive clinical strategies. These strategies should focus not only on the medical suppression of HCV but also on comprehensive fibrosis management and possibly lifestyle interventions that could mitigate the progression of cirrhosis [30,31]. Despite the ledipasvir/sofosbuvir regimen achieving high rates of viral suppression, the development of HCC in 3.30% of the patients and the mortality rate of 5.94% underscore the latent complexities of HCV management. These figures reveal that factors beyond viral presence—particularly fibrosis and baseline liver function—are pivotal determinants of patient outcomes [32,33]. This realization supports the implementation of integrated care models that address both viral eradication and fibrotic control.

Our analysis also brought to light the diminished role of previous interferon treatment in affecting long-term outcomes in the multivariable context, suggesting the interplay of other, possibly genetic or environmental, confounding factors [34,35]. This finding points to the need for broader genomic and epidemiologic studies to discern the varied impacts of past treatments on current HCV therapy outcomes. Age was another significant determinant, with increasing years correlating with rising risks of HCC and mortality. This trend not only highlights the progressive nature of liver disease but also accentuates the cumulative impact of chronic HCV infection, underscoring the importance of early intervention and possibly suggesting benefits from preventative strategies earlier in life [36,37].

The limitations of our observational study, particularly the potential biases from lost follow-ups and unmeasured confounding factors, remind us of the challenges inherent in such extensive longitudinal research. While a proportion of the patients were lost to follow-up, the baseline comparisons showed no meaningful differences in the clinical or biochemical characteristics between those retained and those lost, indicating a minimal risk of selection bias. A slight statistical difference in age was observed; however, the absolute difference was small and not likely to affect the study outcomes. Future studies should aim to incorporate more controlled environments and broader demographic sampling to verify these findings and improve the generalizability of the recommendations [38,39]. Building upon our findings, future research should delve deeper into the longitudinal effects of hepatitis C treatment, particularly examining the sustained impact of therapies like ledipasvir/sofosbuvir over extended periods. Investigating genetic markers that influence the response to treatment could lead to more tailored and effective therapeutic strategies. Additionally, integrated care models that synergize antiviral therapy with interventions targeting lifestyle factors hold promise for improving long-term outcomes in HCV patients. Pharmacoeconomic studies are essential to evaluate the cost-effectiveness of these treatments across different healthcare settings, ensuring that resource allocation maximizes patient benefit. Furthermore, the development of innovative non-invasive techniques for monitoring liver health and fibrosis could significantly enhance the precision and comfort of ongoing patient management. One notable limitation of this study is the lack of precise event dates for outcomes such as HCV RNA relapse, hepatocellular carcinoma (HCC), and mortality. Given the structure of our real-world cohort and the use of annual follow-up visits, it was not feasible to determine the exact timing of these events. Consequently, we applied fixed-time logistic regression models rather than time-to-event methods such as Kaplan–Meier survival analysis or Cox proportional hazards modeling. While this approach limits the temporal resolution of outcome estimation, it reflects the pragmatic nature of real-world longitudinal follow-up in resource-limited settings and is consistent with the methods used in similar observational HCV studies. Future studies with more frequent follow-up or precise event tracking would be valuable to validate and extend these findings.

## 5. Conclusions

This study demonstrates the effectiveness of ledipasvir/sofosbuvir in achieving high rates of initial viral suppression and highlights an encouraging trend of lower viral relapse rates after long-term follow-up. Despite these positive outcomes, the persistent risks of developing hepatocellular carcinoma and mortality, particularly among patients with existing liver fibrosis or cirrhosis, underscore the critical need for ongoing management. These risks remain significant regardless of viral relapse, emphasizing the importance of continuous monitoring and comprehensive care to mitigate the long-term complications.

## Figures and Tables

**Figure 1 viruses-17-00743-f001:**
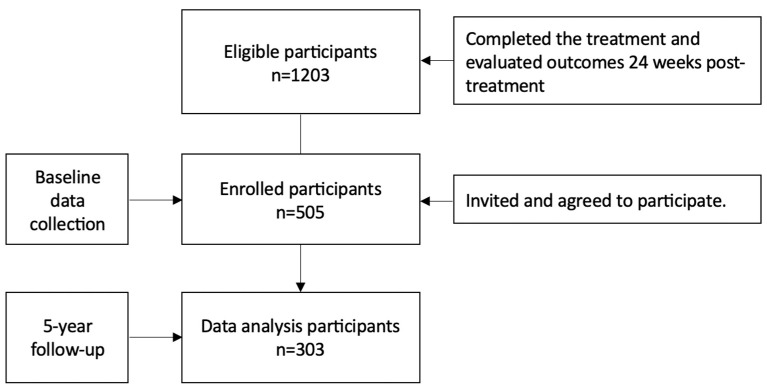
Flowchart of study participants.

**Figure 2 viruses-17-00743-f002:**
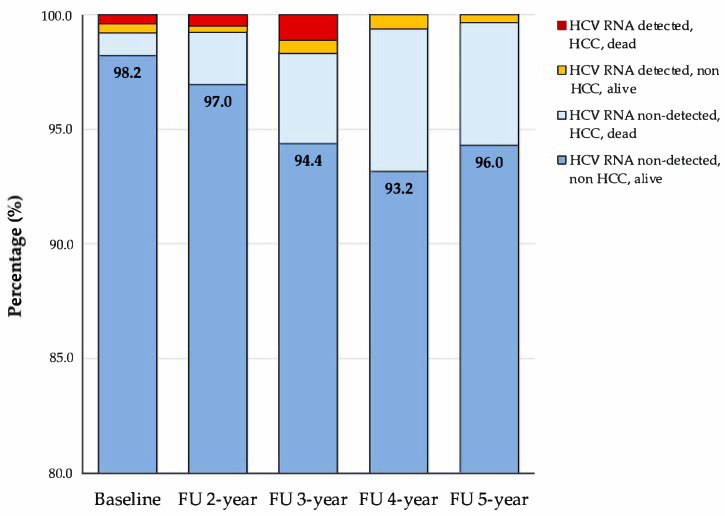
Longitudinal outcomes of HCV treatment over 5 years.

**Table 1 viruses-17-00743-t001:** Longitudinal analysis of HCV RNA detection, hepatocellular carcinoma development, and mortality.

Outcomes	Baseline	FU 2-Year	FU 3-Year	FU 4-Year	FU 5-Year
Enrolled/eligible participants, *n*	505	394	356	322	303
Lost to follow-up rate, %	-	22.0	9.6	9.6	7.5
HCV RNA non-detected, *n* (%)	501 (99.2)	391 (99.2)	350 (98.3)	320 (99.4)	302 (99.7)
HCC non-occurred, alive	99.0 (496)	97.7 (382)	96.0 (336)	93.8 (300)	94.7 (286)
HCC occurred, alive	1.0 (5)	1.8 (7)	2.9 (10)	4.4 (14)	3.6 (11)
HCC non-occurred, dead	0.0 (0)	0.3 (1)	0.3 (1)	0.0 (0)	0.3 (1)
HCC occurred, dead	0.0 (0)	0.3 (1)	0.9 (3)	1.9 (6)	1.3 (4)
HCV RNA detected, *n* (%)	4 (0.8)	3 (0.8)	6 (1.7)	2 (0.6)	1 (0.3)
HCC non-occurred, alive	50.0 (2)	33.3 (1)	33.3 (2)	100.0 (2)	100.0 (1)
HCC occurred, alive	50.0 (2)	66.7 (2)	50.0 (3)	0.0 (0)	0.0 (0)
HCC non-occurred, dead	0.0 (0)	0.0 (0)	0.0 (0)	0.0 (0)	0.0 (0)
HCC occurred, dead	0.0 (0)	0.0 (0)	16.7 (1)	0.0 (0)	0.0 (0)

Data are presented as percentages (numbers).

**Table 2 viruses-17-00743-t002:** Characteristics of study population by HCV RNA detection status.

Findings	HCV RNA Detection		*p*-Value
	Non-Detected	Detected	
Age (years)	52.3 (45–53)	55.1 (45.4–64.7)	0.638
Male sex	99 (34)	4 (40)	0.696
BMI (kg/m^2^)	26.5 (25.9–27.1)	26.8 (23.8–29.8)	0.823
Liver cirrhosis, *n* (%)	49 (16.8)	4 (40)	0.059
Interferon experience	12 (4.1)	1 (10)	
Baseline laboratory tests			
ALT	28.1 (26.2–30.1)	43.8 (14.6–72.9)	0.006
AST	29.1 (26.2–31)	45.2 (21–69.3)	0.007
Total bilirubin	0.60 (0.55–0.67)	1.3 (0.83–1.96)	0.927
GGT	48.4 (43.8–53)	75.2 (30.8–119.6)	0.042
ALP	98.8 (93.1–103.5)	87.8 (51.9–123.5)	0.438
Albumin	42 (41.7–42.4)	40.8 (36.1–44.8)	0.226
PLT	206 (192–220)	160 (135–185)	0.228
HCV RNA (iu/mL)	0	542,000.00 ± 1,210,856.4	0.0001
APRI score	0.45 (0.39–0.51)	1.32 (0.55–2.23)	
>0.5 F0	231 (79.4)	4 (40)	<0.0001
0.51–1.5 F1–2	54 (18.6)	1 (10)	
>1.51 F3–4	6 (2.1)	5 (50)	
FIB-4 score	1.49 (1.37–1.62)	2.21 (1.53–2.94)	<0.001
>1.45 F0	192 (66)	4 (40)	0.006
1.46–3.25 F1–2	83 (28.5)	3 (30)	
>3.25 F3–4	16 (5.5)	3 (30)	

Data are presented as mean ± SD and percentages (numbers).

**Table 3 viruses-17-00743-t003:** Characteristics of study population by hepatocellular carcinoma (HCC) presence.

Findings	HCC Presence		*p*-Value
	Non-Present	Present	
Age (years)	52.5 (51.1–53.8)	63.7 (59.4–64)	0.0001
Male sex	90 (32.5)	11 (57.9)	X^2^ − 0.25
BMI (kg/m^2^)	26.3 (25.8–26.8)	28.3 (25.6–31.7)	0.06
Liver cirrhosis, *n* (%)	37 (13.4)	14 (73.7)	0.001
Interferon experience	10 (3.6)	3 (15.8)	0.43
Baseline laboratory tests			
ALT	27.7 (26.2–29.2)	45.7 (32.9–58.4)	0.001
AST	26.9 (25.2–28.6)	41.5 (28.1–54.8)	0.001
Total/bilirubin	0.62 (0.59–0.65)	0.70 (0.57–0.84)	0.15
GGT	45.4 (41.2–49.5)	79.6 (55.6–103.6)	0.001
ALP	95.5 (90.7–100.4)	120.8 (101.2–140.3)	0.01
Albumin	42.3 (41.9–42.6)	39.3 (37.4–41.2)	0.0001
PLT	208.5 (194–223)	164 (137.9–190)	0.11
HCV RNA (IU/mL)	0	1384.1 ± 3713.5	0.001
APRI score	0.44 (0.38–0.49)	0.69 (0.46–0.90)	0.002
>0.5 F0	225 (81.2)	9 (47.4)	0.002
0.51–1.5 F1–2	47 (17)	8 (42.1)	
>1.51 F3–4	5 (1.8)	2 (10.5)	
FIB-4 score	1.48 (1.36–1.61)	2.86 (1.85–3.87)	0.0001
>1.45 F0	192 (69.3)	3 (15.8)	<0.0001
1.46–3.25 F1–2	75 (27.1)	10 (52.6)	
>3.25 F3–4	10 (3.6)	6 (31.6)	

Data are presented as mean ± SD and percentages (numbers).

**Table 4 viruses-17-00743-t004:** Characteristics of study population by mortality status.

Findings	Mortality		*p*-Value
	Without	With	
Age (years)	52.8 (51.5–54.2)	61.6 (55.8–67.4)	0.002
Male sex	95 (33.3)	10 (55.6)	0.055
BMI (kg/m^2^)	26.4 (25.9–26.9)	27.7 (24.6–30.7)	0.250
Liver cirrhosis, *n* (%)	42 (14.7)	13 (72.2)	0.0001
Interferon experience	11 (3.9)	2 (11.1)	0.170
Baseline laboratory tests			
ALT	28.3 (26.7–29.8)	56.3 (33–79.5)	0.001
AST	27.63 (25.7–29.5)	48.6 (32.1–65.2)	0.001
Total/bilirubin	0.62 (0.59–0.65)	0.8 (0.68–0.98)	0.007
GGT	45.7 (41.8–49.8)	109.2 (69.2–149.1)	0.001
ALP	96.7 (91.8–101.5)	124.9 (95–154.8)	0.07
Albumin	42.2 (41.9–42.5)	38.2 (35.5–40.8)	0.0001
PLT	207.5 (193–221)	151.5 (125.5–171.6)	0.051
HCV RNA (IU/mL)	12,583.50 ± 37,170.44	103,333.33 ± 321,347.61	0.15
APRI score	0.43 (0.39–0.48)	1.37 (0.51–2.22)	0.0001
>0.5 F0	229 (80.4)	6 (33.3)	0.001
0.51–1.5 F1–2	51 (17.9)	6 (33.3)	
>1.51 F3–4	5 (1.8)	6 (33.3)	
FIB-4 score	1.46 (1.33–1.58)	2.63 (2.02–3.24)	0.0001
>1.45 F0	193 (67.7)	3 (16.7)	<0.0001
1.46–3.25 F1–2	78 (27.4)	9 (50)	
>3.25 F3–4	14 (4.9)	6 (33.3)	

Data are presented as mean ± SD and percentages (numbers).

**Table 5 viruses-17-00743-t005:** Univariable and multivariable analysis of factors associated with risk of HCC occurrence.

Variables	Association of Variables with Increased HCC Risk							
	Univariable OR	95% CI		*p* Value	Multivariable OR	95% CI		*p* Value
		Lower Bound	Upper Bound			Lower Bound	Upper Bound	
Age (years)	1.098	1.048	1.151	<0.001	1.081	1.021	1.145	0.003
Liver cirrhosis	18.162	6.179	53.385	<0.001	5.866	1.672	22.577	0.006
Interferon experience	5.006	1.253	20.005	0.023	2.527	0.379	16.863	0.339
ALT	1.033	1.013	1.053	0.001	-	-	-	-
AST	1.056	1.030	1.083	<0.001	-	-	-	-
GGT	1.016	1.007	1.025	0.001	1.011	1.004	1.018	0.001
APRI score	1.893	0.997	3.593	0.051	-	-	-	-
FIB-4 score	2.077	1.434	3.008	<0.001	1.121	0.773	1.626	0.547

Data presented as odds ratio with 95% confidence intervals (95% CI).

**Table 6 viruses-17-00743-t006:** Univariable and multivariable analysis of factors associated with risk of mortality.

Variables	Association of Variables with Increased Mortality Risk							
	Univariable OR	95% CI		*p* Value	Multivariable OR	95% CI		*p* Value
		Lower Bound	Upper Bound			Lower Bound	Upper Bound	
Age (years)	1.071	1.025	1.119	0.002	1.083	1.024	1.147	0.006
Liver cirrhosis	15.043	5.097	44.392	<0.001	6.529	1.913	22.281	0.003
Interferon experience	3.114	0.636	15.249	0.161	-	-	-	-
ALT	1.034	1.017	1.051	<0.001	-	-	-	-
AST	1.050	1.026	1.075	<0.001	-	-	-	-
GGT	1.021	1.012	1.031	<0.001	1.011	1.004	1.017	0.002
Albumin	0.763	0.666	0.874	<0.001	-	-	-	-
APRI score	3.654	1.773	7.529	<0.001	-	-	-	-
FIB-4 score	1.636	1.222	2.189	0.001	1.108	0.764	1.606	0.589

Data presented as odds ratio with 95% confidence intervals (95% CI).

## Data Availability

The data used to support the findings of this study are available from the corresponding author upon request.

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
