# Peer review of "Long-Term Outcomes of Ledipasvir/Sofosbuvir Treatment in Hepatitis C: Viral Suppression, Hepatocellular Carcinoma, and Mortality in Mongolia"

_viruses, 2025, doi:10.3390/v17060743_

Round 1
Reviewer 1 Report
Comments and Suggestions for Authors
This study provides valuable long-term real-world data on the outcomes of ledipasvir/sofosbuvir therapy in a high-prevalence, resource-limited setting, addressing a critical gap in the hepatitis C literature. The study not only confirms the durability of virological suppression but also evaluates clinically significant outcomes such as HCC and mortality. The use of non-invasive fibrosis assessment and routine ultrasound surveillance reflects a pragmatic approach applicable to similar low-resource settings. Importantly, the identification of risk factors for HCC and death—particularly cirrhosis and elevated fibrosis indices—underscores the need for ongoing post-SVR monitoring. These findings contribute meaningfully to our understanding of the long-term clinical impact of direct-acting antivirals and have direct implications for public health strategies in countries with high HCV burdens.
Some points need to be addressed prior to publication.
Major points:
- The study addresses important long-term outcomes of DAA therapy, but given the use of annual follow-up intervals, the analysis of time-dependent outcomes such as HCC and mortality could be improved. While precise event dates may be unavailable, a practical solution would be to approximate the event timing by assigning it to the midpoint between the last negative and the first positive visit. This approach is widely accepted and allows standard time-to-event methods like Kaplan–Meier survival curves and Cox regression to be applied. Incorporating this method would strengthen the robustness of your findings and better reflect the longitudinal nature of the study. If this is not feasible, the manuscript should explicitly acknowledge the limitation of using fixed-time logistic regression for outcomes that evolve over time.
- To manage potential collinearity, a simple and effective approach would be to avoid including both FIB-4 and its component variables (age, AST, ALT) in the same model. Selecting one variable to represent liver fibrosis (e.g., FIB-4 or cirrhosis status) and excluding overlapping variables is a practical way to reduce collinearity while maintaining interpretability of the model.
- In Tables 2a–2c, baseline characteristics are compared between those with and without outcomes. However, the manuscript does not clarify how patients lost to follow-up were handled. I suggest comparing baseline data between retained and lost subjects to evaluate representativeness.
Minor points:
- The manuscript inconsistently refers to the study endpoint as both four-year (lines 72, 173, 179) and five-year (e.g., lines 20, 32, 59) follow-up. This creates confusion. Please standardize this throughout the text and clarify whether the final outcome data reflect four years of follow-up or the full five years as originally intended.
- The manuscript does not define how cirrhosis was diagnosed. Please add a definition in the Methods section, including whether it was based on clinical signs, imaging, non-invasive scores, or a combination thereof."
- The manuscript refers to 'HCV RNA not detected' without specifying whether this corresponds to standard definitions of sustained virological response (SVR12 or SVR24). Please clarify the actual timing of post-treatment viral load assessment and define how virological response was determined.
- The manuscript refers to 'univariate' and 'multivariate' analyses, but the statistical models assess the effect of one or more independent variables on a single outcome. Therefore, the correct terms are 'univariable' and 'multivariable' analyses. Please revise the terminology throughout the text, tables, and figure captions for consistency with standard epidemiological usage.
The English is generally understandable but would benefit from moderate editing to improve clarity, grammar and consistency.
Author Response
Thank you for your valuable comments to improve our manuscript. We appreciate the opportunity to revise and strengthen our work. We have attached our point-by-point responses in the accompanying Word file.

Reviewer 2 Report
Comments and Suggestions for Authors
The study highlights the importance of long-term management to mitigate risks of HCC occurrence and mortality after Ledipasvir/Sofosbuvir initial treatment, especially in patients with severe liver fibrosis or cirrhosis on the backdrop of rather unique circumstances in Mongolia with notably high HCV prevalence in total population. However, if someone considers the population groups that are considered high risk exposure niches, the implications of the authors findings may as well be considered as globally applicable. Therefore, considering the significance of content, scientific soundness and interest to wide range of the readers in virology field, I would recommend this article to be considered to publication in Viruses.
Minor comments
Figure 1 – Thickness of the lanes in flowchart is not uniform, not sure if that was the intention, if not, please correct.
Lane 88 – change “tests was” to “tests were”
Authors need to explain better which cohort of the patients they are meaning, because in some instances the numbers are slightly off. For Example: Lane 168-169 – it says - “Despite this, during the follow-up, 19 patients (6.27%) experienced a relapse of HCV infection”, although in the Table1 the sum adds up to 16 patients with HCV RNA detected.
Table 1, row 3 - it says - “HCV RNA non-detected, % (n)” however, the actual numbers are arranged in opposite way, as “n (%)”.
Figure 2. it is challenging to see small cohorts in the bar graph, I would suggest using different colors instead of the shades of blue.
Author Response

(The authors gave the same response as above.)
